# Snow Leopard (*Panthera uncia*) Activity Patterns Using Camera Traps in the Qilian Mountain National Park (Qinghai Area), China

**DOI:** 10.3390/ani14182680

**Published:** 2024-09-14

**Authors:** Hu Ma, Bading Qiuying, Zhanlei Rong, Jinhu Zhang, Guozhu Liang, Shuguang Ma, Yayue Gao, Shengyun Chen

**Affiliations:** 1Key Laboratory of Surface Processes and Ecological Conservation on the Tibetan Plateau, Ministry of Education, Qinghai Normal University, Xining 810008, China; mh980819@163.com (H.M.); qhnu_palden@163.com (B.Q.); zhanghhjj01@sina.com (J.Z.); 15936237965@163.com (G.L.); 15147954879@163.com (S.M.); 2Qinghai Key Laboratory of Natural Geography and Environmental Processes, School of Geography, Qinghai Normal University, Xining 810008, China; 3Qinghai Administration of Qilian Mountain National Park, Xining 810008, China; gaoyayue0115@163.com; 4Cryosphere and Eco-Environment Research Station of Shule River Headwaters, State Key Laboratory of Cryospheric Science, North-West Institute of Eco-Environment and Resources, Chinese Academy of Sciences, Lanzhou 730000, China; sychen@lzb.ac.cn

**Keywords:** activity patterns, camera traps, coefficient of overlapping, kernel density estimator, Qilian Mountain National Park (Qinghai area), snow leopard

## Abstract

**Simple Summary:**

Understanding the activity patterns of snow leopards is important for the conservation of this species, but the activity patterns of snow leopards in the Qilian Mountain National Park (Qinghai area) are largely unknown. Therefore, we used camera trap data for snow leopards and analyzed activity patterns at different time scales and under different weather conditions. The results demonstrate the patterns of regional snow leopard activity and play an important role in the management and conservation of snow leopards in the region.

**Abstract:**

In recent years, there has been growing concern about the condition of snow leopards. The snow leopard (*Panthera uncia*), an apex predator of alpine ecosystems, is essential for the structural and functional stability of ecosystems. Monitoring of snow leopards’ activity patterns based on camera traps in the Qilian Mountain National Park (Qinghai area) between August 2020 to October 2023 was performed. The results showed that autumn is the peak period of snow leopard activity, especially in September when the frequency of activity is the highest, and there is one peak in the frequency of snow leopard daily activity in the time period of 18:00–22:00, while the highest overlap of the daily activity curves of snow leopards in different months was from spring to autumn (Δ = 0.97), and there were significant differences in diurnal activity rhythm between spring and autumn (*p* = 0.002). Snow leopards prefer sunny days, and they tend to be active at temperatures of −10–9 °C. Our research aimed to uncover the activity patterns of snow leopards at different scales within the study area and provide data for further studies on snow leopards and other wildlife by researchers. This study can be used to gain a comprehensive understanding of the ecological characteristics of snow leopards and to assess their habitats, and it will also serve as a reference for the local wildlife management authorities in formulating snow leopard conservation measures.

## 1. Introduction

The activity regularity of wild animals is an important research topic in ecology, and its patterns have obvious circadian patterns and seasonal differences [1]. The activities of life in nature have various patterns, and animal activity patterns are a regular activity produced by the process of an animal’s long-term adaptation to their natural environment, a composite index subject to a variety of natural conditions, such as the relative activity intensity index. Relative activity intensity is used to evaluate animal activity patterns [2]. Differences in animal activity patterns can intuitively help clarify the composition of an ecosystem and reflects the living conditions and ecological status of important species [3,4].

The snow leopard belongs to the genus *Panthera* of the Felidae, and the snow leopard was listed as a Vulnerable (VU) species by the International Union for Conservation of Nature (IUCN) in 2017 [5], due to various threats that the species faces across its global distribution range. In recent years, there has been an increasing number of studies on snow leopards [6]. These studies mainly involve topics such as snow leopards’ diet types and their food sources [7], species feeding behavior and breeding strategies [8], habitat evaluation [9,10], population density [11], and human–wildlife conflict [12,13]. Also, some studies have focused on the niche of the companion species of snow leopards across spatial and temporal scales [14]. However, due to the complex terrain, inconvenient transportation possibilities, and difficulties in logistics when conducting studies in snow leopard distribution areas, field investigation activities are difficult to implement and few studies have been carried out on snow leopard activity patterns in response to different temperatures. This is particularly true in the Qilian Mountain National Park.

Camera trapping technology has various advantages in wildlife surveys when compared with traditional methods. The technology is better suited for surveying wildlife at high-altitude locations and for conducting continuous monitoring work, and the technology is less invasive than other methods, which is especially suitable for monitoring the activities of hidden and rare animals [15]. Also, it can collect information on the population numbers and the distribution of wildlife [16]. At present, camera trap technology is used to monitor the movements of snow leopards in countries such as Mongolia [17,18,19], Nepal [20,21], India [22,23], Pakistan [24], and Kyrgyzstan [25]. In recent years, researchers have mainly carried out relevant studies in Qinghai [26], Gansu [27], Xinjiang [28], Tibet [29], and Sichuan [30,31]. Our study area has a high elevation and rugged terrain, making the use of camera traps more suitable compared to other snow leopard survey methods.

The Qilian Mountains are one of the most important conservation areas for snow leopards in China. The topics of studies on snow leopards in this region include habitat suitability evaluation [8], human–animal conflict [32], genetic diversity [33], and population density [34]. But studies of the snow leopard’s activity pattern dynamics in the Qilian Mountains are almost nonexistent. We applied to carry out a survey of snow leopard activity in the Qilian Mountains. The objectives of the survey were to comprehensively understand the activity patterns of snow leopards in the region and to analyze the ecological factors affecting these patterns to improve conservation efforts in the region.

## 2. Materials and Methods

### 2.1. Study Area

The Qilian Mountain National Park (Qinghai area) is located at the northeastern edge of the Qinghai–Tibetan Plateau, covering an area of 1.58 × 10^4^ km^2^, which represents 31.5% of the total area of the Qilian Mountain National Park. The average altitude of the Qilian Mountain National Park (Qinghai area) is approximately 4000 m. The diverse ecosystems of the region are an important ecological security barrier for western China. Also, the region is a prioritized area for biodiversity conservation because it harbors unique and rich biological diversity [35]. The average annual temperature is −5.3 °C, with an average annual precipitation of 400 mm, and the climate is characterized by a highland continental climate. There are approximately 41,000 people living in 57 villages and 9 townships in 4 counties (cities). These administrative units include Delingha City, Qilian County, Tianjun County, and the Menyuan County of the Qinghai Province. The Qilian Mountain National Park was established to protect the forest ecosystems, wildlife, glaciers, and wetlands that are distributed in the area [36].

### 2.2. Camera Trapping

Before the installation of the camera traps, we learned about the general activity area of snow leopards in the study area through a survey interview method, and then we set up the cameras according to the local habitat type and the vertical gradient of altitude. A total of 69 camera traps were set up in July 2019 in Huaerdi (elevation 3025~3039 m), Ningzhanggou (elevation 3669~4547 m), Denglonggou (elevation 3983~4543 m), and Wahusi (elevation 3889~4318 m), in Tianjun County, Qinghai Province (Figure 1).

We had two criteria when selecting camera trap sites for installation. Firstly, camera trap sites had to be located in areas with some traces of snow leopard activity (including fecal traces, scratches, spoors, and more) or animal trails; secondly, camera traps were installed at a height of 60~100 cm aboveground, thus avoiding direct sunlight on the camera lens. We also cleared the grass from the filming range, which prevented interference with the normal recording by camera traps.

We used the following camera traps: Model Ltl-6210WMC PLUS, qifone, Zhengzhou, Henan Province, China. The camera modes were set for photo and video to take three consecutive photos and 15 s of video, with a time interval of 1 min, and sensitivity was set to low. The cameras operated 24 h a day. After installing the cameras, the investigators recorded the number of the camera traps, the time of placement, and the elevation of the location. Camera traps were left uninterrupted in the period from August 2020 to October 2023. During the study period, we checked the camera traps every 4 months to replace batteries and memory cards.

### 2.3. Data Analysis

For the same camera traps, we grouped photos into events according to the methods of Michalski [37] and Xue Yadong [38]. If photographs of a snow leopard were taken consecutively within 30 min, the first snow leopard appearance was recorded as a valid photo. A total of 1388 photos and videos of snow leopards were obtained during the monitoring time periods. Of these, 288 photo samples were considered valid for further analysis. We also recorded the time, weather conditions, and temperature at the time of snow leopard occurrence in the cameras.

The relative activity intensity index was used to represent the different days, months, seasons, and temperature intervals of the snow leopard activity time distribution; the calculation formula is [39]:RAI*i* = (T*i*/N*i*) × 100%
where RAI (relative activity intensity index) represents the relative activity of the species, T*i* represents the number of independent photographs in the *i*th time period, and N*i* represents the total number of independent photographs of snow leopards captured by all camera traps.

In plotting the correlation between snow leopard activity and weather conditions, the weather was characterized as one of the following: clear days, when bright sunlight was visible (including visible sunrise and sunset light); cloudy days, when no sunlight was visible and the weather was overcast; or snowy days, when it was snowing (including nights when it was snowing). The percentage of photos for each weather condition was calculated by comparing the number of photos obtained with the total number of weather photos.

We used IBM SPSS Statistics 24.0 (International Business Machines Corporation, 1, North Castle Drive, Armonk, NY, USA) to perform a chi-square test on the relative intensity of snow leopard activity in each of the four seasons. R-4.3.3 was used to map the activities of snow leopards at various scales, in which the overlap package was used to map the daily activity patterns. The daily activity patterns of snow leopards were analyzed based on the kernel density estimation method, and the differences in the activity patterns of snow leopards in different seasons were analyzed according to the overlap of the activity rhythm curves. We used the coefficient of overlapping (Δ) to quantify differences in the activity patterns of this species between seasons in study area. We used Kullback–Leibler divergence to calculate the significance of differences in activity rhythms in different seasons, and the significance level of the test was set to *p* < 0.05. ArcMap 10.8 (Environmental Systems Research Institute, Redlands, California, USA) was used to map the study area.

## 3. Results

### 3.1. Patterns of Snow Leopard Daily Activity

There was one peak and one trough (as opposed peaks) in the daily activity of snow leopards. The peak was present at 18:00–22:00 with a relative activity intensity of 29.71%, and there was a trough at 12:00–14:00 with a relative activity intensity of 3.47%. The relative activity intensity of snow leopards showed an upward trend after 14:00 and began to decline after 22:00. Snow leopard night activity is more concentrated in the summer, and the relative activity intensity index for this period was as high as 63.2% with activity peaking at 22:00–24:00. In contrast, snow leopards are active in the evening twilight earliest in spring(Figure 2).

The highest overlap of the daily activity curves of snow leopards across different months was from spring–autumn (Δ = 0.97) and the lowest overlap of the diurnal activity curve is in spring–winter (Δ = 0.84). There were significant differences in diurnal activity rhythm between spring and autumn (*p* = 0.002), and the significant differences were between spring and winter (*p* = 0.013), autumn and winter (*p* = 0.022), and summer and winter (*p* = 0.028) (Figure 3).

### 3.2. Patterns of Snow Leopard Monthly Activity

Over the course of a year, snow leopards reach their peak activity intensity in September, with a relative activity intensity of 11.81%, followed by a secondary activity peak of 11.11% in October; however, their relative activity intensity decreased from October to November. Snow leopard relative activity intensity increased from November to December. When analyzed from the beginning of a year, relative activity intensity reached an annual low in January to February, but began to show an increase after February, and a peak in activity was observed in May, with a relatively high intensity of activity reaching 10.07%. The relative activity intensity of snow leopards maintained an upward trend from June to September (Figure 4).

### 3.3. Patterns of Snow Leopard Seasonal Activity

Snow leopards are most active in Autumn (December–February), with a relative activity intensity of 29.51%, followed by spring (March–May), summer (June–August), and winter (September–November). Based on the results of the chi-square test for four seasons of the year, there were no significant differences in the relative activity intensity of snow leopards across the four seasons of the year (Figure 5, Table 1, χ^2^ *=* 5.972, *p >* 0.05).

### 3.4. Correlation of Snow Leopard Activity with Temperature and Weather Conditions

The recorded temperatures that correspond to the valid images were categorized according to a gradient of 5 °C and were divided into 14 temperature intervals. The relative activity intensity was lowest in the temperature ranges of −30~−26 °C and 20~≥35 °C. The relative activity intensity increased by 67.01% at temperatures between −6~5 °C, which showed that snow leopards are more active in this temperature range (Figure 6).

We excluded nighttime photos without snow and photos that do not accurately reflect the weather. Of the valid photos considered for data analyses, 43% were taken on clear days, followed by 40.7% taken on snowy days, and the smallest percentage, 16.3%, were taken on cloudy days (Figure 7). This reflects the fact that snow leopards are more active in sunny weather.

## 4. Discussion

We investigated the activity patterns of snow leopards in regard to temperature changes across different temporal scales in the Qilian Mountain National Park. The results of this study complement and improve the literature on snow leopards in the Qilian Mountains and contribute to the implementation of effective measures for snow leopard management and conservation in local protected areas.

The daily activity of snow leopards in the region increases rapidly in intensity from 16:00–18:00 and begins to decline after reaching its peak at 18:00–22:00. In addition, the daily activity patterns across all seasons shows the highest relative intensity of activity at night, which is similar to the activity time of snow leopards in the Wolong Reserve, Sichuan, China [40]. The behavioral patterns of snow leopards in the study area are more frequently nocturnal activity in the summer and activity is concentrated between 20:00 and 4:00, which is similar to the behavioral pattern of snow leopards in southwestern Mongolia [41]. The reason for the lack of significant differences in daily activity patterns between snow leopards in the study area and those in other ranges is because of the universal adaptability of the animals in regard to activity patterns. We found that although there is a significant difference in the diurnal activity of snow leopards between spring and autumn, their activity overlap is high, with multiple time points of activity overlap within a day, especially after 18:00. The discovery of differences in the patterns of snow leopard activity may help reduce niche overlap with other species to some extent, thereby lowering the intensity of interspecies competition. This provides essential scientific evidence for further understanding the coexistence mechanisms of sympatric species and for research on the behavioral ecology of snow leopards.

The results of our study proved that the relative intensity of annual snow leopard activity in Qinghai area reached its maximum in September and then decreased; this was followed by a rebound in December. After February of the following year, snow leopard activity began to show an increasing trend in intensity and reached a sub-peak of activity intensity in May, because snow leopards typically mate from January to March and give birth from April to June [42]. This is in contrast to the annual activity pattern of snow leopards in the Gongga Reserve in Sichuan, which is biased toward a peak activity in May [43]. This variance is due to the low population of snow leopards in the Gongga Reserve, so the circumstances that led to the camera traps being able to capture the snow leopard were fortuitous. Research on the food habits of snow leopards shows that blue sheep are their main prey objects, and understanding the food habits of snow leopards can indirectly help us to understand their seasonal activities and predatory behaviors [6]. The results of the seasonal activity patterns of blue sheep showed a higher frequency of activity in summer and a lower frequency in winter [44]. There are studies proving that blue sheep are one of the most important factors influencing the behavioral activities of snow leopards in the Qilian Mountain National Park, and the increase in the activity intensity of snow leopards may be related to their energy demand as they enter into their breeding season and cub rearing stage. As can be seen through the analysis of seasonal patterns, snow leopards in the study area are most active in autumn. This deviates from the seasonal activity patterns observed in Wolong Reserve in Sichuan, where snow leopards tend to be most active in winter [40]. The study area’s geographical location and higher altitude resulted in the lower autumn temperatures compared to other study areas, leading to an earlier attainment of the temperature range suitable for snow leopard activities. Some studies to prove that climate change may cause the snow leopards’ range to shift upwards or northwards in the future [45,46], and this may affect future changes regarding the habitat range suitable for snow leopards in this study area.

We analyzed the different temperature intervals in which snow leopards appeared, the higher temperature activity intensity index of snow leopards in the study area was concentrated at −10~9 °C. Similarly, the higher temperature activity intensity index of snow leopards in Wolong Nature Reserve was concentrated in the range of −10 °C~−3 °C [40], which shows that snow leopards from different study areas differed in their temperature activity intensity, although there was overlap between them. Comparative studies have found that the altitude of the Wolong Nature Reserve is 3536~4481 m [40], and studies in the central Tianshan Mountains have suggested that the primary distribution area of snow leopards is at altitudes in the range of 2300~3000 m [47]. However, we observed snow leopards at altitudes of 4000~5000 m. Therefore, in addition to seasonal factors, when determining the appropriate temperature range for snow leopards, researchers should consider the optimum ambient temperature range for snow leopards in different areas in relation to the altitude range and temperature range of the area in which they are active. In this study, when we considered weather conditions as impact factors, the highest number of valid photographs of snow leopards were taken on clear days, accounting for 43% of the total number of valid photographs. This indicated that snow leopards preferred sunny weather for their activities. Weather conditions heavily influence animal activity patterns [48], and the most important factor driving feline activity is the activity patterns of their primary prey [49,50]. Several studies have demonstrated that the distributional activities of snow leopards in the Qilian Mountains of Gansu are mainly influenced by prey and altitude [32,51]. Findings from snow leopard surveys in Gansu and Qinghai indicate that key prey items such as blue sheep and Himalayan marmots are important factors influencing snow leopard distribution [52,53]. Blue sheep, as herbivores, have specific activity patterns and preferred weather conditions, and their activity is highest on sunny days as they choose their activities according to different weather conditions [54,55]. The snow leopard’s preference for sunny days is due to the fact that its sympatric prey, especially the blue sheep, tends to be active on sunny days. Also, the results of the weather conditions in which snow leopards choose to operate show a small difference of only 2.3% between sunny and snowy days. During our data collection, the Qilian Mountain National Park did not have any rainy days. The reason for this is that the camera’s elevation is situated near the snow line, resulting in predominantly snowy precipitation. Herbivores expand their feeding range due to the buried food in the snow, which leads to the expansion of the predation activities and range of snow leopards. Therefore, prey activity is important factors influencing which weather conditions snow leopards select to carry out their activities in this study area.

## 5. Conclusions

In terms of daily activity patterns, snow leopards in this study area were most active at night; however, snow leopards have a later activity peak in summer, and they carry out their activities slightly earlier in winter. In terms of annual activity patterns, snow leopards were most active in the fall, especially in September when they reached their peak activity for the year. At the same time, a sub-peak in the intensity of their yearly activity was seen in May. Conservation management authorities should pay attention to the seasons, months, and time periods when snow leopard activity is frequent during routine patrols to avoid disturbing the daily activities of wildlife and should manage human activities to ensure that snow leopards can hunt and breed normally. We found the lowest ambient temperature at which snow leopards were photographed was −27.5 °C and the highest ambient temperature was 36 °C. It was found that their relative activity intensity index peaked at 21.18% at −5~−1 °C, and their relative activity intensity was concentrated in the interval of −10~9 °C, increasing to 67.01%. Due to the influence of temperature and food habits, snow leopards tend to choose sunny days for their activities. Research results show the temperature range and seasonal activity of snow leopards, and in accordance with previous studies, climate change may lead to an upward or northward movement of snow leopards’ range in the future, which may result in changes in the habitat range suitable for snow leopards in the future in this study area. We recommend greater protection for plateau species like the snow leopard, and paying increased attention to the future impact of climate change on snow leopards is not only conducive to their conservation but also conducive to understanding the response of ecosystems to climate change in areas such as the Tibetan Plateau.

## Figures and Tables

**Figure 1 animals-14-02680-f001:**
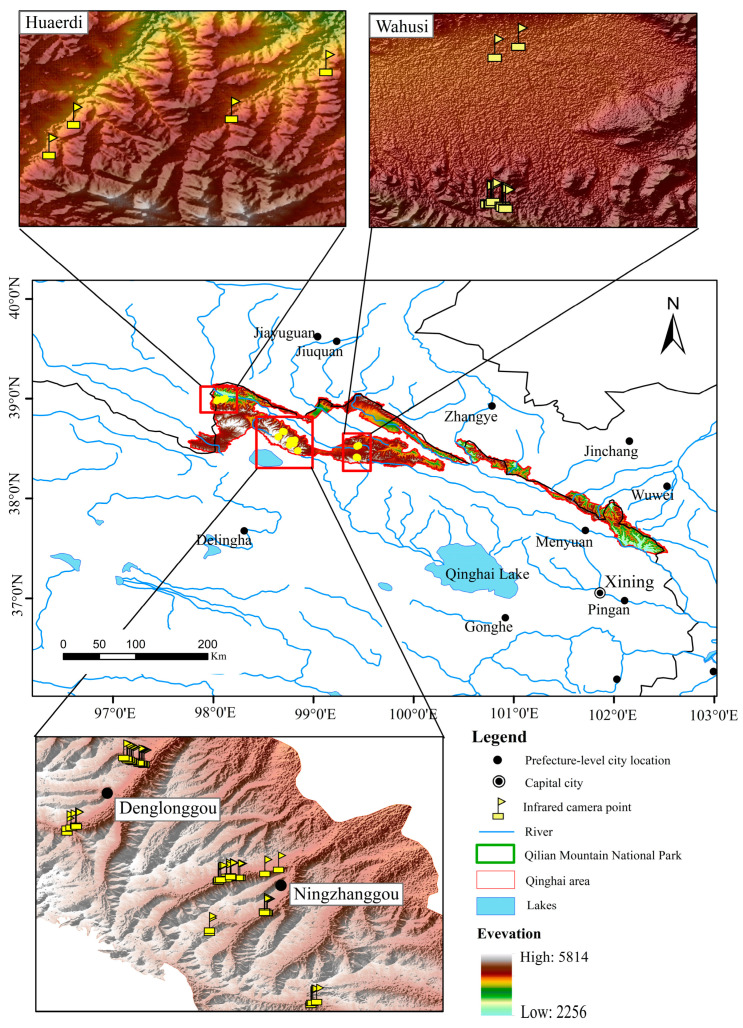
A map of the study area showing the distribution of camera traps for monitoring snow leopard activity.

**Figure 2 animals-14-02680-f002:**
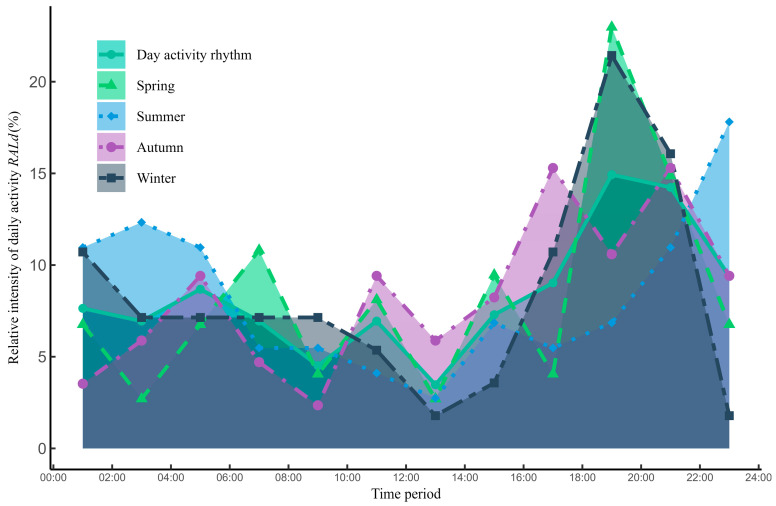
Relative activity intensity of snow leopard daily activity and seasonal variation.

**Figure 3 animals-14-02680-f003:**
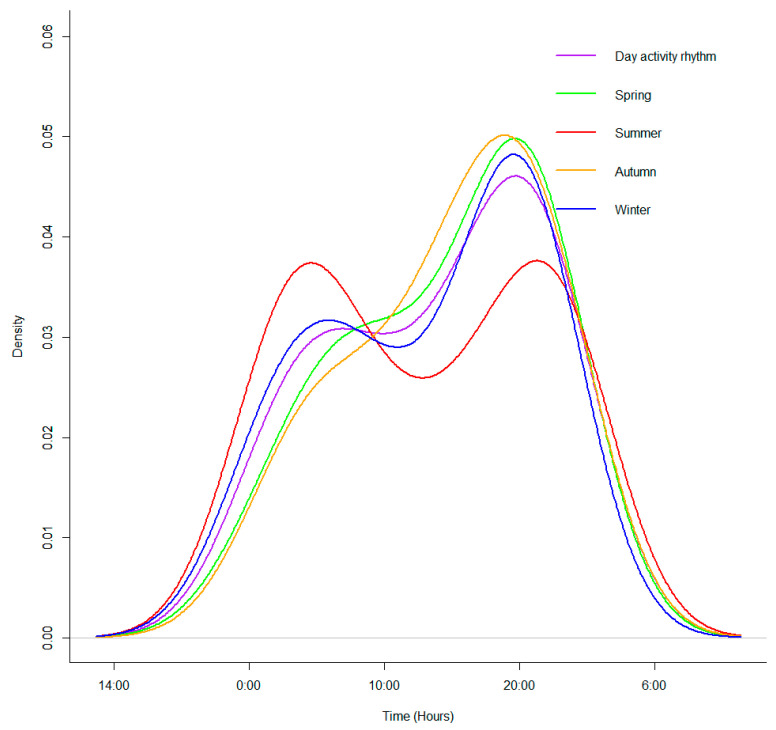
Snow leopard daily activity and daily activity curves across different seasons.

**Figure 4 animals-14-02680-f004:**
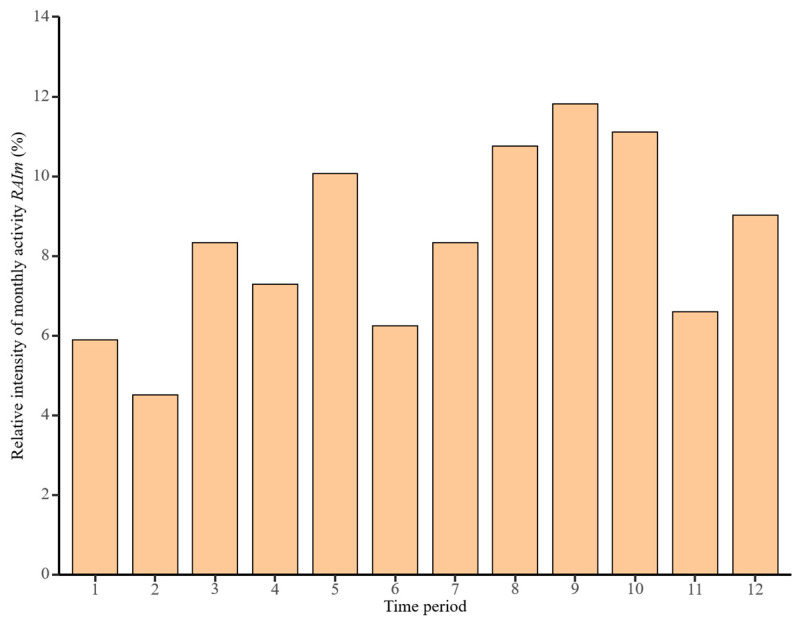
Relative activity intensity of monthly snow leopard activity.

**Figure 5 animals-14-02680-f005:**
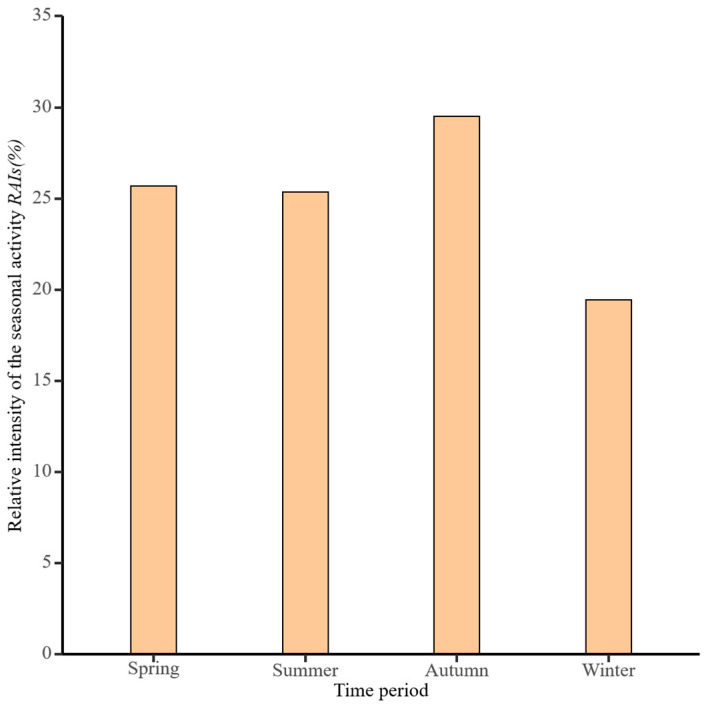
Relative activity intensity of seasonal snow leopard activity.

**Figure 6 animals-14-02680-f006:**
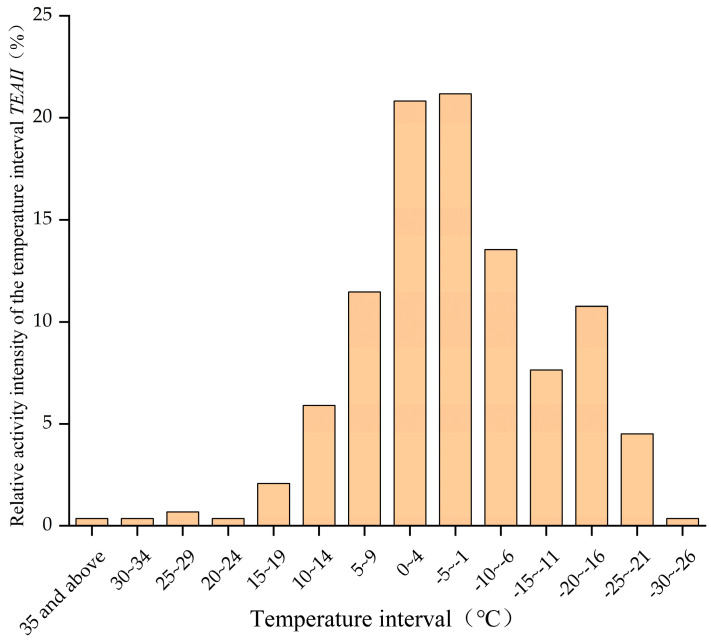
Relative activity intensity of snow leopard activity at different temperature intervals.

**Figure 7 animals-14-02680-f007:**
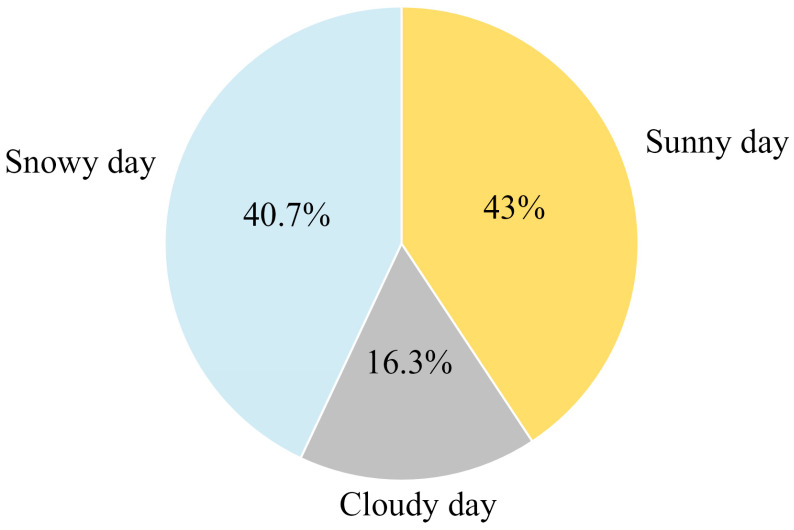
Percentages of valid snow leopard photographs in all weather types. Note: Sunny day—bright sunlight visible (includes visible sunrise and sunset light); cloudy day—sunlight not visible and weather is overcast; snowy day—weather in which snow is falling (includes nighttime when snow is falling).

**Table 1 animals-14-02680-t001:** The chi-square test for the relative intensity of snow leopard activity during the four seasons of the year.

Season	Number of Valid Photographs	Relative Intensity of Activity (%)	Results
Spring	74	25.69	*χ^2^* = 5.972
Summer	73	25.35	*p* > 0.05
Autumn	85	29.52	
Winter	56	19.44	

## Data Availability

The data presented in this study are available on request from the corresponding author. The data are not publicly available due to privacy and ethical restrictions to protect the confidentiality of the study participants.

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
