# Peer review of "Snow Leopard (Panthera uncia) Activity Patterns Using Camera Traps in the Qilian Mountain National Park (Qinghai Area), China"

_animals, 2024, doi:10.3390/ani14182680_

Round 1

Reviewer 1 Report

Comments and Suggestions for Authors

Dear Authors

An article submitted for review entitled Research on snow leopard (Panthera uncia) activity rhythms using camera-traps in the Qilian Mountains National Park (Qinghai area), China in its current form does not appear to be sufficiently interestingly written to be published in the journal Animals mdpi. The authors, use common tools-infrared trap cameras-to experiment, studying the behavioral behavior of the title species. Undeniably, behavioral studies on representatives of wild animals are, as a rule, extremely interesting, but here the authors focused only on the analysis of seasonal , daily activity, without giving an answer to what specifically "activity" of these animals they write about, so the reader reads about the activity that is more or less depending on the seasons and nothing comes out of it ... In addition, the team of researchers analyzing the recordings of cameras selectively treats the information, using only those that are, according to the authors of the experiment more useful, such as better recorded. I suggest the authors to improve this work  for readers.

Regards

Author Response

请参阅附件

Reviewer 2 Report

Comments and Suggestions for Authors

The following issues should be fully addressed to accept for a publication in the journal.

1. Introduction:  The introduction sets a concrete background and correctly identifies research gap. It would strengthen the manuscript to include a brief discussion on how understanding these snow leopards’ activity patterns might impact specific conservation practices in the Qilian Mountains.

2. Methods: The description of the camera trap methods and the data collection processes are adequately detailed. However, the manuscript should clarify why the chosen camera locations and settings (e.g., camera height, mode, and sensitivity) were optimal for meeting the objectives of the study.  The statistical methods used are appropriate but the authors should provide the rationale for choosing these methods over other potential analytical approaches.

3. Results:  The results are detailed and informative. However, the presentation could be improved by consistently using the same terms for time periods across all figures and tables to avoid confusion (e.g., twilight vs. dusk).  The section on temperature and weather correlations is compelling; consider adding a visual representation of snow leopard activity across the temperature gradients to enhance interpretability.

4. Discussion:  The discussion effectively ties the findings back to the literature, but also tends to generalize the implications. It would benefit from a more detailed exploration of how these patterns of activity could affect interactions with prey species, particularly in different weather conditions.  The manuscript mentions the adaptation of snow leopards to their environment briefly. Expanding on this point by discussing potential evolutionary advantages or survival strategies inferred from the activity data would enrich the discussion.

5. Conclusion:  The conclusion aptly summarizes the findings. Authors could enhance this section by explicitly stating how this research can be applied by park managers or in conservation strategies.  The authors should provide suggestions future research directions that address the limitations noted, such as a larger sample size or additional environmental factors like snow depth.

6. Figures and Tables:  Ensure that all figures and tables are cited in the text and discussed sufficiently to justify their inclusion. Consider improving the resolution and clarity of the graphical data, particularly for Figure 4, where the activity peaks could be highlighted or annotated for better visibility.

7. References:  The references are relevant and sufficiently current. Adding a few more recent studies, especially from 2021-2023, could provide a more robust context for the findings.

8. Recommendation:  This manuscript should be considered for publication after moderate revisions that address the specific points noted above. The study contributes valuable baseline data on snow leopard activity that could significantly impact local conservation practices.

Comments on the Quality of English Language

The manuscript is generally well-written. Minor grammatical errors should be corrected for professional presentation.

Reviewer 3 Report

Comments and Suggestions for Authors

Comment:

Line 33 and 34: "The snow leopard, scientifically classified as Panthera uncia, is a member of the genus Panthera within the Felidae family. The term 'subfamily Leopardidae' is incorrect in the context of snow leopards."

Line 43: "These studies mainly focus on topics such as the snow leopard's diet. For further information, refer to https://www.mdpi.com/2076-2615/13/20/3182 and https://www.frontiersin.org/articles/10.3389/fevo.2021.783546/full”

Line 51 - 64: "The paragraph is not essential in the introduction section and should be removed."

Map: "The map requires revision for improved clarity of the digital elevation model (DEM)."

IR camera model: "Detailed information regarding the infrared (IR) camera model used in the study is necessary."

Transect line survey: "It is unclear why and how the authors used a transect line survey method."

Data on weather conditions: "The manuscript lacks clarity on how data regarding weather conditions, temperature, and other information at the time of snow leopard occurrences in the cameras were collected. If this was based on temperature records from camera traps, it requires validation, as camera trap temperature records may not always match ambient temperatures."

Camera trap deployment: "It is not clear how the camera traps were deployed and the rationale behind their deployment."

Total number of snow leopard photos: "The manuscript should specify the total number of photos (events) of snow leopards recorded. Furthermore, are these photos sufficient for daily activity analysis? Refer to https://link.springer.com/article/10.1007/s00265-015-1929-6  for further details."

Graph development and calculation: "It is suggested to use the R package 'overlap' for the development of graphs and calculation of the relative intensity of snow leopard daily activity. Refer to https://zslpublications.onlinelibrary.wiley.com/doi/full/10.1111/j.1469-7998.2011.00864.x, https://link.springer.com/article/10.1007/s00265-015-1929-6 and https://www.sciencedirect.com/science/article/pii/S2351989424001574#bib11 for guidance."

Figure 3 and Table 1: "The results presented in Figure 3 and Table 1 may not be valid analyses, as the seasonal variation in the times of sunset and sunrise can affect the activity patterns of snow leopards. Check https://zslpublications.onlinelibrary.wiley.com/doi/full/10.1111/j.1469-7998.2011.00864.x and https://doi.org/10.25225/fozo.v66.i4.a4.2017 for further insights. Revision of the analysis based on the variation in sunshine and sunset times is recommended."

Figure 4: "Figure 4 is not suitable for joining lines in different months. It should be presented as either a bar diagram or in another suitable format, but not as a line graph."

Time demarcation for seasons: "The manuscript lacks clarity on the demarcation of time for spring, summer, autumn, and winter."

Overall revision: "The manuscript requires a major revision of the methods and result section. The result may vary after following the valid methodology for activity pattern. Furthermore, I am afraid that the English of this MS is appalling and sometimes understanding it is not an easy job: it must be revised (actually, re-phrased!) thoroughly to make it understandable. Overall, I like the manuscript. But for this to get published it needs a good deal of rewriting.

Comments on the Quality of English Language

English of this MS is appalling and sometimes understanding it is not an easy job: it must be revised (actually, re-phrased!) thoroughly to make it understandable. 

Author Response

请参阅附件

Reviewer 4 Report

Comments and Suggestions for Authors

This manuscript presents an interesting study of “Research on snow leopard (Panthera uncia) activity rhythms using camera-traps in the Qilian Mountains National Park (Qinghai area), China”.

It investigates the activity rhythms of snow leopards in the Qilian Mountains National Park (Qinghai area), in western China. The data for the current study was obtained from camera-traps installed in the study area. Followed by analysis of the activity pattern on different time scales, the pattern of its activity in temperature intervals and the weather preferences of its activity. This study was conducted in time span of three years (from August 2020 to October 2023). The result revealed that autumn is the peak period of snow leopard activity, especially in September when the frequency of activity is the highest, and there is one peak in the frequency of snow leopard daily activity in the time period of 18:00 - 22:00, and the relative intensity of activity can reach 46.88% at night. Importantly, the Snow leopards prefer sunny day for their activities, and their relative activity intensity can reach 67.01% at temperatures ranging from -10 ~ 9 ℃. The activity pattern of snow leopards reflects their adaptation to the environment, and the monitoring data and activity rhythms of infrared cameras can provide important scientific value for the management and conservation of snow leopard in this area of its distribution.

This paper is organized, has clear objectives and the drawn conclusions are coherent with the obtained results. The results are well presented and the conclusions are relevant for environmental conservation community. However, there are some aspects of the manuscript that I believe should be addressed by the authors before publication. Firstly, the text needs editing to correct typographical, grammatical and spelling mistakes. Examples are many to be included in this report, but I could select a few just from the manuscript:

1.     The scientific names throughout the manuscript (starting from the title) should be italicized.

2.     The key-words should be alphabetically arranged.

3.     I think it is better to use abbreviation for these terms, for example Qilian Mountains National Park is may be abbreviated as “QMNP”. Similarly Qinghai-Tibetan Plateau may be abbreviated as “QTP” as observed in the published literature. I think this will help in easy understanding for the readers.

4.     The citations style is different throughout the manuscript. I suggest to use uniform citation style in the manuscript.

5.     Line # 39 to Line # 41, Please revise this sentence as “In 2017, the snow leopard was listed as a Vulnerable (VU) species by the IUCN [5], due to various threats the species faces across its global distribution ranges”.

6.     Line # 42, I suggest to add a reference to this sentence as, “In recent years, there has been an increasing number of researches on snow leopards (Rashid et al. 2020) ”.

·       Rashid W, Shi J, Rahim Iu, Sultan H, Dong S, Ahmad L. 2020. Research trends and management options in human-snow leopard conflict. Biological Conservation 242:108413. https://doi.org/10.1016/j.biocon.2020.108413.

7.     Line # 98 to Line # 99, Please check this sentence and revise it, “sensitivity was set in the middle of the range”.

8.     In Table 1 and Table 2, please check the caption and also use a uniform font size in the tables.

9.     Please check the sub-heading in the section 4.4 and revise it accordingly.

10.  Please revise the caption of Figure5. Seasonal relative activity intensity of snow leopards at Qinghai Part of Qilian Mountains National Park (Qinghai area). It seems that the name Qinghai is duplicated here.

11.  Line # 226, Please correct the spellings in this sentence (Qinhai area).

12.  Line # 245, Please remove the Error! Reference source not found..  in this sentence.

13.  Line # 274 to Line 276 , Please revise this sentence as “and studies in the central Tianshan Mountains have suggested that the hotpot distribution area of snow leopards is in the range of 2300 ~ 3000 m [35]”.

14.  Line # 280 to Line 282 , Please revise this sentence as, “In this study, when considered weather conditions as impact factors, the highest number of valid photographs of snow leopards were taken on clear days, accounting for 43 % of the total number of valid photographs”.

15.  Line # 324 to Line 325, Please remove this sentence, “Please turn to the CRediT taxonomy for the term explanation. Authorship must be limited to those who have contributed substantially to the work reported.

16.  Line # 328 to Line 350, Please remove these paragraphs from the manuscript.

17.  Last but not the least, Please use a uniform style and format for the references. Currently, some of the journals names are written in full, while others are in abbreviated form. For example reference number #27, Similarly the reference # 7 is in different format.

Comments on the Quality of English Language

Minor English language editing required

Round 2

Reviewer 1 Report

Comments and Suggestions for Authors

The authors made an effort to answer the reviewer's questions. I accept the authors' explanations and express a positive opinion on the acceptance of the manuscript in its present form.

Author Response

Dear reviewer: We are very grateful for your professional review of our article.

Reviewer 3 Report

Comments and Suggestions for Authors

No comment

Comments on the Quality of English Language

Looking forward the correction as reply by author 
